# Artificial Sweeteners in Breast Milk: A Clinical Investigation with a Kinetic Perspective

**DOI:** 10.3390/nu14132635

**Published:** 2022-06-25

**Authors:** Sofie Stampe, Magnus Leth-Møller, Eva Greibe, Elke Hoffmann-Lücke, Michael Pedersen, Per Ovesen

**Affiliations:** 1Department of Gynaecology and Obstetrics, Aarhus University Hospital and Steno Diabetes Centre Aarhus, 8200 Aarhus N, Denmark; magnusmoeller@clin.au.dk; 2Comparative Medicine Laboratory, Aarhus University, 8000 Aarhus, Denmark; michael@clin.au.dk; 3Institute for Clinical Medicine, Health, Aarhus University, 8000 Aarhus, Denmark; evagreib@rm.dk (E.G.); elkehoff@rm.dk (E.H.-L.); 4Department of Clinical Biochemistry, Aarhus University Hospital, 8200 Aarhus N, Denmark

**Keywords:** breastfeeding, breast milk, artificial sweeteners, offspring health, lactation, infant, nutrition

## Abstract

Artificial sweeteners (ASs) are calorie-free chemical substances used instead of sugar to sweeten foods and drinks. Pregnant women with obesity or diabetes are often recommended to substitute sugary products with ASs to prevent an increase in body weight. However, some recent controversy surrounding ASs relates to concerns about the risk of obesity caused by a variety of metabolic changes, both in the mother and the offspring. This study addressed these concerns and investigated the biodistribution of ASs in plasma and breast milk of lactating women to clarify whether ASs can transfer from mother to offspring through breast milk. We recruited 49 lactating women who were provided with a beverage containing four different ASs (acesulfame-potassium, saccharin, cyclamate, and sucralose). Blood and breast milk samples were collected before and up to six hours after consumption. The women were categorized: BMI < 25 (*n* = 20), BMI > 27 (*n* = 21) and type 1 diabetes (*n* = 8). We found that all four ASs were present in maternal plasma and breast milk. The time-to-peak was 30–120 min in plasma and 240–300 min in breast milk. Area under the curve (AUC) ratios in breast milk were 88.9% for acesulfame-potassium, 38.9% for saccharin, and 1.9% for cyclamate. We observed no differences in ASs distributions between the groups.

## 1. Introduction

The prevalence of obesity worldwide has increased tremendously [1], leading to increasing interest in low-calorie alternatives to sugar [2]. Artificial sweeteners (ASs), or sugar alternatives, can replace sugar in foods, drinks, and other products, and they are chemical additives with a strong sweet taste and a non-existing or extremely small caloric load. A large American cross-sectional study reported an intake of ASs in 41.4% of all participating adults [3]. The frequency of ASs consumption was increased with body mass index. Moreover, a considerably larger intake was reported in individuals with diabetes (67.4%) than in individuals without diabetes (39.0%). 

Although ASs minimize the calorie content of a product, there is some ongoing controversy over whether ASs usage poses health risks. Potential adverse effects of ASs have been investigated, and several suggestions have been made: a high ASs exposure may lead to weight gain [4], increased food intake by adjusting satiety [5], increased preference for sweet foods [6,7], altered glucose intolerance [8], reduced insulin sensitivity [9,10], and altered gut microbiome by suppression of valuable bacteria [11] and by upregulation of inflammatory cytokine pathways in subcutaneous adipose tissue [12]. Additionally, two follow-up studies suggested a relationship between maternal intake of ASs during pregnancy and the offspring’s risk of obesity at the age of 1 year [13] and 7 years [14]. Collectively, these factors contribute to a higher risk of weight gain in people ingesting ASs and concerns about the offspring’s health later in life [15] following AS ingestion during pregnancy. 

Nevertheless, health authorities and general practitioners often recommend that pregnant women, who have diabetes and/or are obese, substitute regular sugars with ASs to prevent a significant weight gain during pregnancy, gestational hypertensive disorder, and gestational diabetes [16]. Accordingly, ASs seem to be popular in pregnant women, and in a study including 59,334 Danish pregnant women, more than one-third reported intake of artificially sweetened beverages during pregnancy [17]. Recent studies provided evidence of human transplacental transmission of ASs [7]. While ASs have been detected in maternal breast milk in both humans [18,19] and animals [7], there is a lack of information about the mechanisms related to ASs in breast milk. 

Thus, the aim of this study was to investigate the pharmacokinetic properties of different types of ASs (acesulfame-potassium (ace-k), saccharin, cyclamate, and sucralose) in plasma and breast milk after oral administration in lactating women. Furthermore, we aimed to investigate the potential dependencies of ASs’ biodistribution on weight and metabolic status (normal-weight, overweight, and women with diabetes). Lastly, we aimed to relate the accumulated plasma-to-milk transfer of ASs to the standard-acceptable daily intake (ADI) recommendations. 

## 2. Materials and Methods

### 2.1. Subjects

To investigate the pharmacokinetic properties of ace-k, saccharin, cyclamate, and sucralose in breast milk, we conducted an open-label, prospective clinical trial. Women were divided into three groups: normal weight (BMI < 25), overweight women (BMI > 27), and women with type 1 diabetes mellitus (T1DM). Each woman was weighed on the day of participation. The remaining information was obtained from electronic patient records (Table 1). Inclusion criteria were defined as the age above 18, ability to deliver breast milk samples, and BMI < 25 or BMI > 27. The women were not required to be exclusively breastfeeding to participate. A statement of consent was obtained by each woman before parti-cipation. The protocol was approved by the National Committee on Health Research Ethics before recruitment (protocol code 1-10-72-219-20. 28 October 2020). ClinicalTrials.gov ID: NCT-04578431.

### 2.2. Study Design

The women were instructed to avoid all ASs at least 24 h before participation. Furthermore, they fasted overnight prior to participation; however, women with diabetes were allowed to eat or drink to prevent potential hypoglycaemic episodes, which are likely to happen in lactating women with diabetes [20,21]. Upon arrival, baseline blood and breast milk samples were collected. Then, each woman consumed a beverage created by us containing 200 mL water, 30 mL unsweetened cranberry juice for taste, and 10 mL AS concentrate containing 85 mg ace-k, 75 mg sucralose, 60 mg cyclamate, and 20 mg saccharin. Consumption of the drink was labelled as time = 0 min (t = 0). In total, 8 blood samples and 8 breast milk samples were collected at times t = 0, 30, 60, 120, 180, 240, 300, and 360 min. The women were provided with three standardized meals without ASs (breakfast, a snack, and lunch) throughout the 360 min and were not allowed to eat anything else during the study. This was to ensure that the women did not ingest any other ASs than those provided in the beverage at baseline. 

Blood samples were collected in 4 mL EDTA tubes via a peripheral venous catheter placed in the elbow area. Breast milk samples were collected via a manual breast pump (Harmony; Medela, McHenry, IL 60050, USA) by the women themselves. A minimum of 4 mL of breast milk was collected per sample. The same breast was not necessarily used for every sampling. The women were allowed to breastfeed their offspring throughout the study period, and therefore the breasts were not necessarily completely emptied throughout the samplings. The breast pump parts were cleaned between each sample collection to remove the remains from the previous sample. Blood samples were centrifuged at 3026× *g* for 10 min. Plasma was pipetted for storage. Plasma and breast milk samples were kept on ice, then stored in a −80 °C freezer until analysis. Samples were analyzed all at once at the Department of Clinical Biochemistry at Aarhus University Hospital, Denmark. 

### 2.3. Biochemical Analyses

AS concentrations were measured by High-Performance Liquid Chromatography with Tandem Mass Spectrometry (LC-MS/MS). The liquid chromatography was carried out on an Agilent 1290 Infinity Series system (Agilent Technologies, Glostrup, Denmark), and mass spectrometric detection was carried out on an Agilent 6470 Triple Quad mass spectrometer (Agilent Technologies, Glostrup, Denmark), equipped with an electrospray ionization source. Analytical separation was performed on a Luna Omega C18 column (2.1 × 50 mm, 1.6 µm) (Phenomenex, Copenhagen, Denmark) at a temperature of 30 °C controlled by a column heater. The method is described in detail by Greibe et al. [22].

### 2.4. Statistics

Statistical analyses and graphs were created using GraphPad Prism (version 9.2.0 for Mac OS X, GraphPad Software, San Diego, CA, USA, www.graphpad.com, accessed on 1 August 2021). Furthermore, it was used to define the lack of normal distribution in our data using QQ-plots. 

Study data were stored in REDCap (Research Electronic Data Capture) at Aarhus University [23,24]. 

## 3. Results

Recruitment and execution of the study took place from November 2020 to June 2021. In total, 60 women signed up to participate. However, only 49 completed the study as the remaining failed to participate because of sickness in either mother or offspring upon participation or insufficient milk production. 

We found that all four sweeteners were present in maternal plasma and breast milk after oral ingestion. All values are median ± interquartile range (IQR) since our data are not normally distributed. Outliers were calculated for each sweetener. An outlier was identified as three standard deviations from the mean [25]. Thus, three women were excluded for ace-k, three for saccharin, five for cyclamate, and three for sucralose (Table 2). These were all different women except for two; one who presented as an outlier in ace-k, saccharin, and cyclamate, and one who presented as an outlier in ace-k and sucralose. The outlier was only removed from the sweetener in which she presented as an outlier, and she was removed from both plasma and breast milk data sets. 

Figure 1 shows concentrations of the ASs over time in plasma and breast milk. These graphs have been used to determine the area under the curve (AUC) ratios. The AUC ratio determines how much sweetener is transferred from plasma to breast milk in percentages. However, AUC ratios were calculated before the sweeteners reached complete elimination, as none of the sweeteners did during the 360 min study period. Thus, the AUC ratios are relative to the time of 360 min. The lower limit of quantitation (LOQ) of the sucralose analyzing method was 10 ng/mL. Therefore, we cannot report a peak concentration of sucralose in breast milk as it never reached above 10 ng/mL, nor could the AUC ratio be calculated for sucralose. Peak concentrations, time, and AUC ratios are presented in Table 2. Moreover, it shows the total number of participants in each group after outliers were removed.

We observed no visual difference in plasma or breast milk concentration levels of any of the sweeteners between the three groups. We did not perform any statistical test as the graphs clearly illustrate no significant difference (Figure 2).

Lastly, Table 3 illustrates the acceptable daily intake (ADI) established by the European Food Safety Authority (EFSA) [26]. We have calculated how many of our beverages a woman can drink per day to reach the ADI for herself directly and her offspring through breast milk. This index is calculated for each sweetener, and these varied markedly between the four sweeteners. For ace-k, a woman should drink 8 of our beverages to attain ADI for herself and 14 for her offspring through breast milk. For saccharin, a woman should drink 19 for herself and 81 for her offspring, and for cyclamate, 9 for herself and 5444 for her offspring. For sucralose, she would drink 15 to attain her own ADI, but because of the lack of AUC ratio for this sweetener, we cannot determine the number of beverages needed to attain the offspring’s ADI. The max daily dose for the woman is based on the average weight of all women participating (76 kg). Moreover, calculations were based on the blood volume being 5320 mL (=70 mL/kg in adult women [27]) and the plasma volume being 3192 mL (assuming the average hematocrit for adult women is 40% [28]). 

## 4. Discussion

Our findings reveal that after oral ingestion of ace-k, saccharin, cyclamate, and sucralose, all are transferred to plasma and then to breast milk in lactating women. Previous studies have confirmed the presence of ace-k, saccharin, and sucralose in breast milk [18,19]; however, cyclamate has not been investigated in breast milk before, presumably because of the fact that cyclamate is banned by the Food and Drug Administration (FDA). However, there is no evidence that cyclamate should be harmful to use [29]. Cyclamate is approved for use by the European Food Safety Authority (EFSA) and is found in various daily products. This study also demonstrates that cyclamate can transfer into breast milk, however, in small amounts (AUC ratio = 1.9%). Interestingly, we found very different AUC ratios among the sweeteners, meaning that the amount of sweetener transferred from plasma to breast milk differs substantially between sweeteners. Note the very low amount of cyclamate and the undetectable amount of sucralose compared with the relatively high amount of ace-k (Table 2). The available literature cannot explain these differences, and there is a need for more research on the kinetics of ASs, but perhaps the varying polarity of the sweeteners can explain this difference [30]. Moreover, we cannot know if the sugars in the cranberry juice of the beverage affect the metabolism of the sweeteners. However, we used unsweetened cranberry juice to minimize the amount of sugar added; only 1.6 g of sugar was found in the cranberry juice per beverage.

As shown in Table 3, a woman must drink between 8 and 19 of the beverages created for this study to reach her acceptable daily intake (ADI) of sweeteners. To reach the offspring’s ADI dose through breast milk, a woman must drink between 14 and 5444 beverages. Note that the quantity of sweeteners in the beverage given in this study exceeds the regular amount found in light products. In a study by Rother et al., participants were given a beverage containing 68 mg sucralose and 41 mg of ace-k [19]; almost the same quantity of sucralose and around half of the quantity of ace-k found in our beverage. An 8 oz can of Diet Coca-Cola contains 40 mg sucralose and 30 mg ace-k, whereas Coca-Cola Zero contains 31 mg of ace-k [31], equivalent to around half of the amount of sucralose and almost a third of ace-k in our beverage. Thus, there are more sweeteners in our beverage than in usual products containing ASs.

A limit to the calculation in Table 3 is that we cannot make sure that the ADI is applicable to infants as well. However, to our knowledge, these data are the only ones to calculate from. Moreover, it is important to notice that this study does not conclude that intake of ASs below the ADI is healthy or advisable for either mother or nursing offspring. 

Another popular sweetener, aspartame, is used in over 6000 products [32,33], but we chose not to include it in this study as it is rapidly metabolized in the gut to phenylalanine, aspartic acid, and methanol [34]. It is therefore difficult to detect aspartame in plasma and breast milk. However, the use of aspartame might have the same health-related issues as implied for the other sweeteners, and the health status of aspartame is likewise highly debated [33]. 

We obtained no information about the women’s habitual consumption of ASs. Dietary questionnaires do not necessarily provide a clear picture of ASs intake because of the risk of recall bias. Furthermore, ASs are found in many products without the consumer’s knowledge [18]. ASs are not only found in food products but also in chewing gum, medications, hygiene products, etc. [35]. This widespread use might explain some of the outliers, where sweeteners were found in plasma and breast milk at baseline. Moreover, the outlying values can be due to a lack of compliance or uncertainty in measurements. To our knowledge, habitual consumption or abstention does not influence the distribution of ASs in the body, and, therefore, questionnaires have been left out of this study. 

Our results indicated that there is no difference between the three groups of women, suggesting no relationship between weight and metabolic status and the distribution of ASs in plasma and breast milk. We consider these findings positive as people with diabetes tend to use ASs in larger amounts compared with people without diabetes [3].

The strengths of the study include a wide variety of sweeteners and several repeated measures. Moreover, both plasma and breast milk samples have been obtained for comparison, which makes calculating AUC ratios possible. However, it should be noted that these AUC ratios are calculated before the complete elimination of the sweeteners has been reached. Therefore, the AUC ratios are relative to the 360 min post-ingestion. AUC ratios may look different if calculated after complete elimination, but this was not possible in our study. Unfortunately, it was not possible to calculate the AUC ratios for sucralose as a majority of the samples fell below the LOQ of the analytical method [22]. 

Though repeated measures were obtained, it would have been interesting to obtain samples for longer than 360 min to determine when the sweeteners are completely washed out from plasma and breast milk. However, no study has, to our knowledge, obtained samples of ASs in blood and breast milk as long in duration as ours, and this is considered a strength of this study.

The literature suggests that sweeteners (ace-k and saccharin) are quickly and almost completely absorbed from the gastrointestinal system after oral ingestion [30,36]. Sweeteners are excreted through the kidneys into urine after absorption, and the unabsorbed sweeteners are excreted unchanged through feces. A future study could investigate the fate of sweeteners in the human body for more than six hours.

This study examined the pharmacokinetic aspects of ASs in women exclusively. Future studies could investigate the offspring’s concentration of ASs in plasma after being breastfed by their mother following maternal ingestion of ASs. As exclusive breastfeeding is recommended for six months, followed by partial breastfeeding until 12 months of age or longer [37,38], clinical studies are warranted to determine whether early exposure to ASs via breast milk may have clinical implications.

## 5. Conclusions

Ace-k, saccharin, cyclamate, and sucralose all transfer into plasma and breast milk, and they can be detected several hours after oral ingestion. The time for peak concentration was between 30–120 min in plasma and 240–300 min in breast milk. However, very large amounts were required to reach maximum daily intake (ADI) in both mother and offspring. Moreover, pharmacokinetic aspects of ASs were independent of the weight or metabolic status of the mother ingesting them. Lastly, we found that AUC ratios varied considerably depending on the sweetener, as AUC ratios in breast milk were 88.9% for ace-k, 38.9% for saccharin, and 1.9% for cyclamate.

## Figures and Tables

**Figure 1 nutrients-14-02635-f001:**
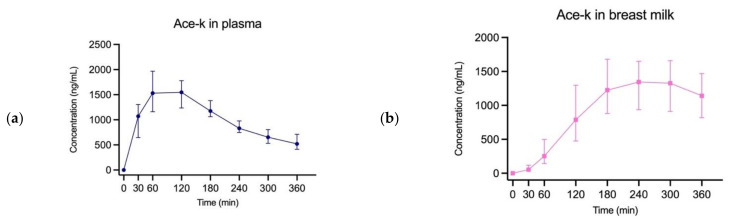
Median concentrations with IQR of sweeteners in plasma (to the left, (**a**,**c**,**e**,**g**)) and breast milk (to the right, (**b**,**d**,**f**)). Concentration of sucralose in breast milk did not reach LOQ and therefore no graph could be created. Outliers have been removed for each sweetener: a + b) *n* = 46, c + d) *n* = 46, e + f) *n* = 44, g) *n* = 46.

**Figure 2 nutrients-14-02635-f002:**
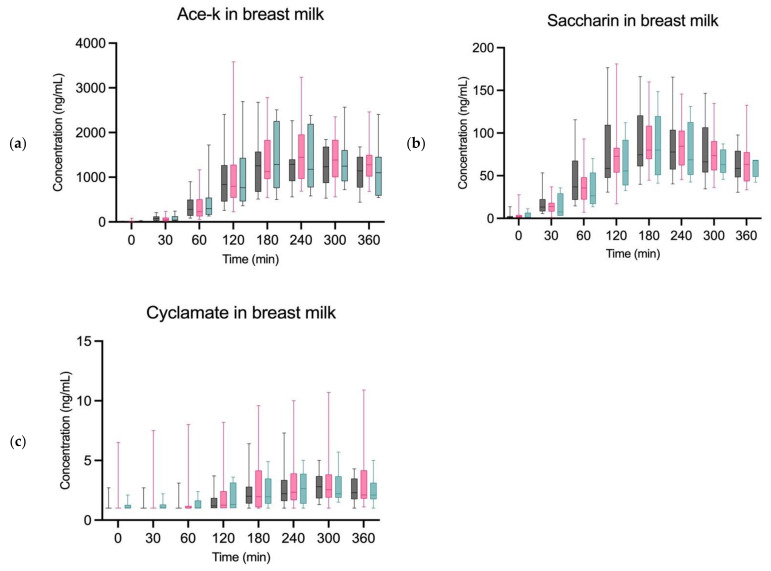
Grouped concentrations of ASs in breast milk. (**a**) represents grouped ace-k in breast milk, (**b**) represents grouped saccharin in breast milk and (**c**) represents grouped cyclamate in breast milk The dark grey represents group 1 (BMI < 25), the pink represents group 2 (BMI > 27), and the blue represents group 3 (T1DM). Outliers have been removed for each sweetener.

**Table 1 nutrients-14-02635-t001:** Demographic characteristics of the women and their offspring.

	All	BMI < 25	BMI > 27	T1DM
Women, *n*	49	20	21	8
BMI, kg/m^2^	25.9 (7.6)	21.3 (2.9)	28.8 (6.4)	25.3 (7.7)
Age, years	29 (3)	28 (2)	31 (4)	28.5 (6)
Parity, number	1 (3)	1 (0)	2 (1)	1 (1.8)
Offspring birth weight, g	3780 (508)	3662 (504)	3855 (545)	3981 (417)
Offspring birth length, cm	53 (3)	52.5 (3)	53 (3)	52 (2.8)
Offspring GA, days	218 (17)	282 (14)	288 (17.5)	268 (16.2)
Offspring age, days	121 (154.5)	179.5 (123.5)	94 (141.5)	77.5 (84.5)

Data presented as median (interquartile range (Q3–Q1)). T1DM—type 1 diabetes mellitus, BMI—Body mass index, GA—gestational age at birth. “Age” is the age upon the day of participation.

**Table 2 nutrients-14-02635-t002:** Peak concentration, time of peak, and area under the curve (AUC) for each sweetener.

	*n*	Plasma		Breast Milk		
		Peak Concentration	Time to Peak	Peak Concentration	Time to Peak	AUC Ratio
Ace-k	46	1548 ng/mL	120 min	936 ng/mL	240 min	88.93%
Saccharin	46	350.7 ng/mL	30 min	81.5 ng/mL	240 min	38.91%
Cyclamate	44	160.6 ng/mL	60 min	2.56 ng/mL	300 min	1.86%
Sucralose	46	134.6 ng/mL	120 min	<LOQ	-	-

**Table 3 nutrients-14-02635-t003:** Acceptable daily intake (ADI) according to EFSA.

	Ace-K	Saccharin	Cyclamate	Sucralose
ADI (mg/kg/day) [26]	9	5	7	15
Dose given (mg)	85	20	60	75
Woman max doses (n)	8	19	9	15
Offspring max doses (n)	14	81	5444	-

“Dose given” is the amount of ASs found in the beverage we have created for the study. This could not be calculated for sucralose as the AUC ratio could not be calculated.

## Data Availability

Not applicable.

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
