# Peer review of "Artificial Sweeteners in Breast Milk: A Clinical Investigation with a Kinetic Perspective"

_nutrients, 2022, doi:10.3390/nu14132635_

Round 1

Reviewer 1 Report

In the manuscript titled '"Artificial Sweeteners in Breast Milk: A Clinical Investigation 2 with a Kinetic Perspective"' by Stampe et al., the authors tested whether several artificial sweeteners (AS: acesulfame-potassium, saccharin, cyclamate, and sucralose) can transfer from mother to offspring through breast milk. The analyses of AS concentrations in maternal plasma and breast milk showed that all four AS were detected in maternal plasma and breastmilk. The temporal alterations in the plasma and breastmilk concentrations of each AS have a specific pattern. The authors also analyzed AS concentrations in the samples of three groups of mothers (BMI<25, BMI>25, and the patients with type 1 diabetes mellitus). The results showed that the pharmacokinetic feature of AS was independent of the mothers' body weight or metabolic status.

   The large consumption of AS in modern Western diets has created a great controversy about the risks to human health. However, mothers' risks for AS consumption during the perinatal period have not been discussed well so far. Although the results of this study are not very robust, the author's attempts to reveal the transfer of AS via breast milk are very important from the point of perinatal neonatology and the studies of perinatal environmental factors affecting the development of offspring. The experiments are well-organized and clearly explained, and the results are solid and carefully described. The data that the authors obtained were deliberately mined and sufficiently discussed. Although the manuscript is rather descriptive and does not provide any mechanistic insights, the reviewer believes that accumulating such descriptive data would finally lead to important clinical implications. However, a few issues need to be addressed and clarified.

1.     The authors describe the temporal alterations of the AS in the plasma and the breast milk samples shown in Fig. 1. Then, the authors refer to the AUC ratios in Table 2. Next, the authors show Figure 3, which partially overlaps with the contents of Fig. 1. The AUC ratios are calculated based on the data shown in Fig. 3. This flow of the description of the result section looks a little back and forth. The reviewer requests the authors to combine Fig. 1 and Fig. 3, describe the results shown in these figures at first, and the authors should mention the AUC ratios calculated based on the graph in Fig. 1. The description would become more straightforward so that the readers can easily follow. Moreover, Table 2 should be shown after Figure 1 in the manuscript.

2.       In the current manuscript, Table 3 appears before Figure 3. Table 3 should be shown in the final (and summarizing) part of the manuscript because this estimation is based on the observatory data shown in Figs.1-3. Then, table 3 should be shown in the last part of the result section.

3.     In Figs 2 and 3 in this manuscript, the different groups are illustrated just by colors. For color-blind researchers, different groups should be illustrated with both different colors and different patterns. For example, the data in plasma should be illustrated as blue-solid lines, and the data from breast milk should be shown as pink-dotted lines in Fig. 3. If necessary, explanatory notes (graph keys) should be shown in these graphs so that the readers can tell easily.

Author Response

Dear reviewer. Thank you very much for your feedback and your comments. We hope that you accept the changes we have made. We have addressed your comments in the following:

  1. Please see line #145-154. Furthermore, Figure 3 has been deleted.
    Moreover, Table 2 is now shown after Figure 1

  1. Figure 3 is deleted, so this solves the issue on Figure 3 before Table 3.

  1. Regarding the colors: Colors in Figure 2 have been changed to a color package called “Color Blind Safe” in Prism.

Again, thank you very much for your feedback. All changes made can be seen in the revised manuscript.

Reviewer 2 Report

This manuscript describes a study of breastfeeding mothers given a single dose of juice spiked with four artificial sweeteners followed by blood and breastmilk collection for 6 hours to assess the levels.  The authors calculate the dose of artificial sweeteners received by the infant.  Data is further stratified by BMI and diabetes status.  

The study is significant since women are often told to limit their sugar intake for health reasons and no rigorous study of transfer of artificial sweeteners to infants through human breast milk has been done.  Studies from animals show effects on offspring metabolism and microbiome.  

Whether the breasts were emptied during the 6 hour experiment and whether the mother was exclusively breastfeeding are details that would aid in the interpretation of the data.  Milk sampling should be better described.

This study is largely descriptive.  I hope the authors follow up with a publication stressing the importance of their findings and the need for further research in this area, especially in terms of mechanism of transfer. 

Author Response

Dear reviewer. Thank you very much for your feedback and your comments. We hope that you accept the changes we have made. We have addressed your comments in the following:

  1. The women were not exclusively breastfeeding. Nor were their breasts completely emptied throughout sampling as they had to breastfeed their offspring during the study. This has been added in line #74-75 and #99-103

  1. Milk sampling has been further described, please see line #99-103.

Again, thank you very much for your feedback. All changes made can be seen in the revised manuscript.

Reviewer 3 Report

Thank you to submit the manuscript "Artificial Sweeteners in Breast Milk: A Clinical Investigation

 with a Kinetic Perspective" to Nutrients. The manuscript investigated the concentration of some sweeteners in the plasma of women with obesity or diabetes and also in breast milk from these women. The results obtained in this work are interesting, the article is well written and the subject is within the scope of this journal, however I have a few suggestions.

Lines#34-36: Did the study report a high number of consumption by obese individuals or the general population? My question is what is the justification for using obese individuals in this work and this justification would happen if the study reported in these lines has also reported that the highest percentage of sweetener consumption has occurred in this public.

Lines#86-88: It seems to me that a control group of unsweetened cranberry juice has not been added. Could this juice impact the metabolism of other sweeteners? It is necessary to consider that juices have a high fructose content. Has the composition of this juice been verified? If yes, please add.

Line#124: please bring Figure 1 before Table 2 for consistency with the text

Lines#194-197: it would be interesting to add the possible negative effects reported in studies with dosages lower than the maximum recommendation.

Author Response

Dear reviewer. Thank you very much for your feedback and your comments. We hope that you accept the changes we have made. We have addressed your comments in the following:

  1. Please see line #223-226 where further information regarding the cranberry juice has been added. We chose to use an unsweetened cranberry juice with low amounts of sugar (=1.6 g pr. Beverage) to lower the risk of the sugars interacting with the artificial sweeteners.

  1. Line#34-36: The study did report higher consumption of AS in obese and diabetic individuals. Moreover, many health care professionals tend to recommend both obese and diabetics to substitute sugary product with artificially sweetened ones, especially during pregnancy. This justifies the investigation of normal weight, obese and diabetics in this study.

  1. Table 2 is now shown after Figure 1

Again, thank you very much for your feedback. All changes made can be seen in the revised manuscript.